# Environmental Consciousness and Green Customer Behavior: The Moderating Roles of Incentive Mechanisms

**Chung-Te Ting [1] , Chi-Ming Hsieh [2], Hsiao-Ping Chang [3,4] and Han-Shen Chen [4,5,*]**

[1]   Department of Tourism, Food and Beverage Management, Chang Jung Christian University, Tainan 71101, Taiwan; ctting1973@gmail.com
[2]   International Bachelor Program of Agribusiness, National Chung Hsing University, Taichung 40227, Taiwan; hsiehch9@nchu.edu.tw
[3]   Department of Health Diet and Industry Management, Chung Shan Medical University, Taichung 40201, Taiwan; pamela22@csmu.edu.tw
[4]   Department of Medical Management, Chung Shan Medical University Hospital, Taichung 40201, Taiwan
[5]   Department of Health Diet and Industry Management, Chung Shan Medical University, Taichung 40201, Taiwan
*   Correspondence: allen975@csmu.edu.tw; Tel.: +886-4-2473-0022 (ext. 12225)

**Abstract:** Consumer awareness of environmental protection and energy conservation concepts has prompted businesses in the hotel industry to adopt green operations. Most studies of the hotel industry have discussed the behavioral intentions (BIs) of consumers based on the theory of planned behavior (TPB), but they have not considered emotional and motivational factors. The present study incorporated two incentive mechanisms and the anticipated positive and negative emotions of consumers into the TPB to explore the relationship between BIs and green hotel development. Structural equation modeling was applied to test the research hypotheses. The results indicate that (1) a positive correlation exists between environmental attitude, subjective norms, perceived behavioral control (PBC), positive anticipated emotion, and desire intention (DI), and a negative correlation exists between negative anticipated emotions and DI; (2) a positive correlation exists between PBC, DI, and BI; and (3) an incentive mechanism has a moderating effect on the relationship between DI and BI.

**Keywords:** sustainable consumption; green purchase behavior; corporate green strategies; corporate environmental responsibility; sustainable hotel practices

## 1. Introduction

Environmental problems, such as global warming, are a continually growing concern worldwide [1,2]. Because of the increase in environmental awareness and carbon reduction trends, green hotels have become a prominent business direction in the hospitality industry. Hotels that adopt ecologically friendly designs and operations by adopting environmental practices, such as lowering pollutant emission, procuring green goods, and conserving water and energy, are deemed to be green hotels [3,4]. According to Ecomall [5], green hotels should strive to reduce their effect on the environment through measures such as energy conservation, waste reduction, and water conservation. Green Mountain State [6] reported that green hotels create a natural and healthy environment, thereby encouraging their employees and guests to scrutinize each of their operational procedures for the minimization of environmental impact. Krakovsky [7] argued that hotels should appeal to the environmental awareness of their guests by urging them to reuse towels and refrain

from using disposable amenities during their stay. By adopting these measures, hotels contribute to environmental protection and sustainable development while attracting guests who identify with these concepts [8,9]. Relevant studies have addressed various environmental issues within the hotel industry, including green marketing [10,11], everyday environmental behavior and concerns [12–14], the willingness to pay for green consumption [15,16], and consumer intention to engage in eco-friendly behavior by staying in green hotels [17–22]. Chen and Tung [23] adopted the theory of planned behavior (TPB) to examine the consumer intentions driving the decision to stay at a green hotel. Lorenz et al. [24] used the TPB to explore the intention of consumers to buy products with origin labels, and Yadav and Pathak [25] applied the TPB to study the green purchasing behavior of consumers in developing nations. Wong et al. [26] utilized the extended TPB to study the attitudes and purchase intentions of consumers regarding suboptimal food products.

The failure of the TPB to account for emotional and motivational factors has created a research gap. As Hecken and Bastiaensen [27] indicated, positive external incentives can increase intrinsic motivation, which explains why certain hotels sponsor and promote environmental activities that enhance consumer participation. To fill this research gap, the present study incorporated positive anticipated emotions (PAEs), negative anticipated emotions (NAEs), and incentive mechanisms (cash discounts and eco-friendly substitutes) into the TPB to construct an extended TPB for determining the behavioral intentions (BIs) of consumers toward staying in green hotels. The purpose of this study is twofold: to explore the main reasons that consumers opt to stay in green hotels and determine which incentive mechanism (cash discounts or eco-friendly substitutes) is more effective in enhancing the BIs of consumers to stay in green hotels.

Related studies have primarily utilized the TPB as a basis for consumer BIs, paying little attention to the intrinsic motives of consumers. By contrast, the present study integrated PAEs and NAEs into the TPB to establish an extended TPB for investigating the BIs of consumers to staying in green hotels. This study adopted the two incentive mechanisms (cash discounts and eco-friendly substitutes) of the Environmental Protection Administration's "Join Us! Let's Go Green!" program as moderators and verified that these incentive mechanisms can moderate consumer desire intentions (DIs) and BIs to stay in green hotels by strengthening the positive effects of consumer BIs to select such hotels. The findings of this study might serve as a reference for green hotels to develop strategies that are attractive to guests.

The rest of the paper is arranged as follows: Section 2 covers the literature on TPB, PAEs, and NAEs, presents the hypotheses, and highlights the relationships among the hypotheses. Section 3 explains the research methodology, which encompasses data collection, model construction, and measurement. Then, Section 4 presents the data analysis process, including structural equation modeling (SEM). Next, Section 5 discusses the research and the managerial implications that flow from the findings. Finally, limitations and suggestions for future research are expressed in the Section 6.

## 2. Literature Review

### 2.1. Green Hotel

The green hotel concept originated in Germany in the 1980s. There are quite a few names for describing green hotels, such as "Eco-Hotel" or "Ecological Hotel", "Eco-efficient Hotel", "Eco-friendly Hotel", "Environmentally Friendly Hotel", and other terms. The Green Hotel Association [28] defines a green hotel as follows: "The hotel itself has environmental performance. The hotel management method must actively focus on saving energy, water resources and reducing waste, and avoiding the loss and waste of resources to maintain the global environment. Green hotels usually implement environmental protection measures to reduce the negative impact on the environment, and environmentally responsible practices (ERPs) contain sustainable environmental standards, regulatory orders, best environmental protection measures, environmental protection labels, and environmental management systems (EMSs) and environmental indicators" [29]. Taiwan's Environmental Protection

Administration (EPA) defines green hotels and characterizes their practices thus: "Disposable products reduction, including whether hotels promote sheet reduction, reduce the frequency of towel replacement, and provide disposable toiletries such as toothpaste and toothbrush", resource recycling, etc. [30].

The Green Hotel Association [28] classifies green hotel certifications according to their recycling systems, equipment, ingredients, customers, communities, water resources, information, and item recycling. In terms of the hotel's tours, attractions, and accommodations (including hotels, resorts, permanent camps, camps, and travel trailers), various products are certified by the hotel based on whether they meet environmental standards. The United Kingdom established the Green Tourism Business Scheme (GTBS) in 1997, which provides a guiding assessment for the tourism industry, enabling operators to provide high-quality services and sustainable operations. The tourism industry criteria are divided into tiers: Going Green, copper, silver, and gold. The Hotel Association of Canada (HAC) and the Canadian environmental labeling agency Terra Choice founded and launched the "Audubon Green Leaf Eco-Rating Program" in 1998. Providers of accommodation facilities and services of all sizes—large international tourist hotels, business hotels/restaurants, small and medium-sized motels, holiday cabins, homestays, etc.—can apply to participate in the rating system. The North American Green Leaf Hotel certification program includes water quality, water conservation, waste minimization, resource conservation, and energy efficiency. Its assessment indicators include energy efficiency, conservation of resources, pollution prevention, and environmental management. The content ranges from equipment energy efficiency, indoor air quality, and water conservation to environmental policy and communication. In 2003, the China Hotel Association established the "Green Restaurant Rating Regulations", which focus on safety, health, and environmental protection and are the basis of evaluating and grading hotel companies. The safety aspect focuses on fire safety, public security, food safety, consumer safety, and occupational safety; the health component emphasizes green rooms, green catering, and sanitation operations; the environmental protection element includes clean energy production, attention to energy conservation, consumption, and waste disposal. In response to the 2008 Green Olympics, China actively promoted this certification system and divided the "China Green Hotel" into five levels, from A to AAAAA, where AAAAA is the highest level. In 2012, Taiwan's EPD divided the environmental protection hotel rank into three levels: Gold, Silver, and Copper. The review criteria can be divided into environmental protection policies, energy conservation, water conservation, waste reduction, green procurement, and hazardous materials management. Certain labeling can be obtained by businesses in the tourism industry so that those with environmentally friendly hotel labels can be identified as an "eco-friendly hotel" [30].

Hsieh et al. [31] asserted that enterprises need to innovate and introduce green factors, but the environmental problems encompass such a wide range of issues that enterprises must reform again. Hence, in order to respond to these circumstances, enterprises must implement supporting measures to improve green innovation. In response to external environmental pressures, companies are required to carry out environmental protection work, and this related work is called "green innovation". Chiou and Pan [32] indicated that future green innovation will lead to a competitive advantage and environmental management, and innovation will become an important performance indicator. Businesses practicing green innovation will make it more difficult for their competitors to enter the market, implement green management and green production of their internal products, and gain an overall competitive advantage. Kularatne et al. [33] proposed that green hotels' environmental sustainability in energy and water management helps to increase their competitiveness, saves them money, and attracts environmentally concerned customers. Taking the tourist hotel as an example, the "Tempus Hotel Taichung" launched "Health and Healthy Meal" on the basis of "Reducing Food Miles" by purchasing fresh ingredients to cook, shortening traveling distance, and implementing environmental carbon reduction. On Earth Day, Shangri-La's Far Eastern Plaza Hotel, Taipei, and Evergreen Laurel Hotel and Windsor Hotel, Taichung, encourage employees to ride bicycles to work, avoid elevators, and turn off the lights for 1 h at night [30].

### 2.2. Theory of Planned Behavior (TPB)

The TPB posits that three factors underlie behavioral intentions (BIs): attitudes toward a specific behavior, subjective norms (SNs), and perceived behavioral control (PBC) [34]. Here, "attitude toward a specific behavior" refers to the "degree to which a person has a favorable or unfavorable evaluation or appraisal of the specific behavior", whereas an SN is the "perceived social pressure to perform or not perform the behavior". PBC is defined as "an individual's perceived ease or difficulty of performing the particular behavior". Thus, individuals who perceive themselves as possessing high-level behavioral control for a specific behavior are more likely to exhibit a strong intention to undertake that behavior. The term "behavioral intention" itself is defined as a person's inclination to undertake a specific behavior; the assumption is that this inclination necessarily occurs or is strengthened immediately before the behavior is acted out [34].

### 2.2.1. Environmental Attitude and DI

Environmental attitude (EA) refers to the collection of beliefs, DIs, and BIs a person holds regarding environmental activities or concerns [35]. According to Hirose [36], environmentally friendly behavior is determined by goal intentions (general attitudes) and BIs (intentions to perform certain actions). Goal intentions are influenced by perceived environmental risks (a sense of crisis caused by environmental pollution), environmental responsibilities (believing that taking action to save the environment from pollution and destruction is necessary), and the effectiveness of measures (believing that taking appropriate measures can solve environmental problems). By contrast, BIs are influenced by assessments of feasibility (whether people have adequate knowledge or skills to evaluate environmental friendliness), social norms, and cost versus profit. Hence, the following hypothesis is proposed:

**H1:** *EA has a significant positive effect on the DI of consumers to stay in green hotels.*

### 2.2.2. Anticipated Emotions and DI

Consumer research is related to properties of emotions [37]. Loewenstein and Lerner [38] pointed out that emotions directly and indirectly influence consumer decisions. The emotions reflect the psychological aspects of an individual's expectations for staying in green hotels. The PAEs of consumer significantly affected customers' DI [39]. Hence, we predicted that the PAEs of travelers would significantly influence their DI to stay in green hotels.

**H2:** *PAEs have a significant positive effect on consumer DI to stay in green hotels.*

In addition, Carrus et al. [40] and Han and Yoon [1] suggested that increasing positive emotions and suppressing negative emotions combine to form the most effective strategy for maintaining intentions. The NAEs of consumer significantly affected customers' DI [41,42]. Hence, we predicted that the NAEs of travelers would significantly influence their DI to stay in green hotels.

**H3:** *NAEs have a significant negative effect on consumer DI to stay in green hotels.*

### 2.2.3. SNs and DI

SNs refer to an individual's perception of the beliefs of others regarding whether he or she should perform a certain behavior. SNs often lead to social pressure from family members (intrinsic considerations) or individuals or groups outside of an individual's immediate family (extrinsic considerations). Therefore, relatives and friends possess the ability to encourage sustainable behavior or discourage unsustainable behavior. Hence, the following hypothesis is proposed:

**H4:** *SNs have a significant positive effect on consumer DI to stay in green hotels.*

### 2.2.4. BIs and DI

DI is a mental state that drives an individual to fulfil or realize personal motives; numerous studies have recognized DI as the antecedent driving the BIs of consumers [43–46]. A BI involves an individual's decision regarding whether to perform a certain action after assessing various factors; DI is an effective predictor of BIs [47]. According to Perugini and Bagozzi [45], DI mediation can improve the prediction of BIs. Hence, the following hypothesis is proposed:

**H5:** *The DI to stay in green hotels has a significant positive effect on consumer BIs.*

### 2.2.5. PBC and BIs

PBC refers to an individual's subjective assessment of how easy it is to successfully perform a certain behavior. Han and Yoon [1] pointed out perceived behavioral control has a significant impact on DI; thus the following hypothesis is proposed:

**H6:** *PBC has a significant positive effect on consumer DI to stay in green hotels.*

With reference to Litvine and Wüstenhagen [48], who indicated that PBC governs sustainable behavior, in the consumer's green hotel visit intention, PBC contributes to better prediction of customers' intentions to visit a green hotel [19,49,50]; thus, the following hypothesis is proposed:

**H7:** *PBC has a significant positive effect on consumer BI to stay in green hotels.*

### 2.2.6. Relationships of Incentive Mechanisms with DI and BIs

An incentive mechanism compels an individual to work toward the accomplishment of a certain goal. In management science, the most common incentives that drive employees to work hard, other than a sense of achievement and recognition, are monetary and material incentives. Consumption incentive mechanisms are a policy or plan by governments or companies that encourage consumers to increase their purchase intention. Frey and Jegen [51] and Frey and Oberholzer-Gee [52] argued that extrinsic monetary intervention can weaken intrinsic motivation. Therefore, cash discounts (high cash discounts or low cash discounts) were selected to examine their moderating effects on the link between DI and BI. Thus, the following hypothesis is proposed:

**H8a:** *The incentive mechanism of cash discounts affects the DI and BIs of consumers regarding the decision to stay in green hotels.*

Moreover, Luo et al. [53] indicated that the environmental impact has pressured governments to implement subsidy incentives to promote the electric vehicle. Zhang et al. [54] studied consumer purchases of electric vehicles with incentive mechanisms in place, and the result shows that whether the adoption of subsidization is sustainable largely depends on consumers' purchase intentions. Therefore, eco-friendly substitutes (a high portion of funds or low portion of funds) were selected to examine their moderating effects on the link between DI and BI. Accordingly, the following hypothesis is proposed:

**H8b:** *The incentive mechanism of eco-friendly substitutes affects the DI and BIs of consumers regarding the decision to stay in green hotels.*

## 3. Methodology

### 3.1. Research Framework

This study selected the TPB as its basis. Incorporating two additional aspects, PAEs and NAEs, into the TPB resulted in a comprehensive model of the TPB. The present paper describes the opinions and perspectives of Taiwanese consumers regarding green hotels and details their purchase DIs. Figure 1 presents the proposed framework.

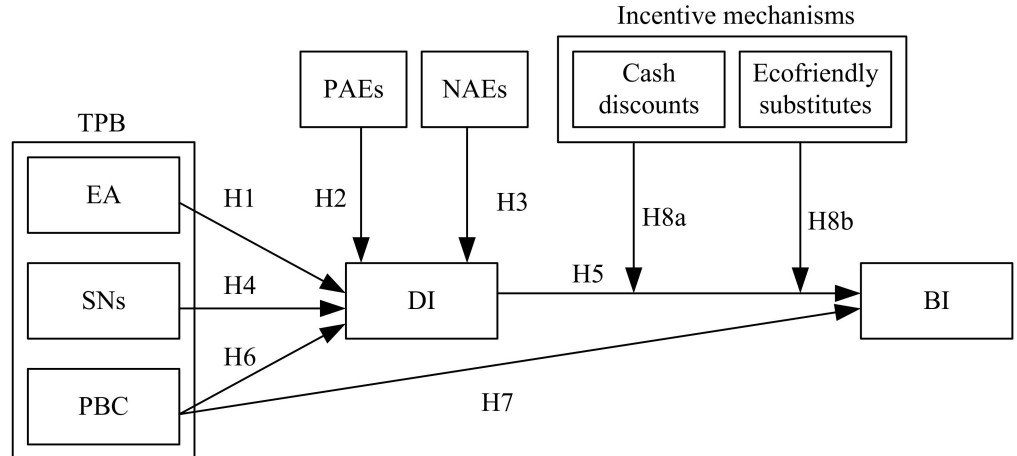

TPB: theory of planned behavior; EA: environmental attitude; SNs: subjective norms; PBC: perceived behavioral control; H: hypothesis; PAEs: positive anticipated emotions; NAEs: negative anticipated emotions; DIs: desire intentions; BI: behavioral intention.

**Figure 1.** Research framework.

## 3.2. Questionnaire Design

The design of questionnaire items stemmed from a review of pertinent literature and involved the use of a seven-point Likert scale, with 7 representing "strongly agree" and 1 representing "strongly disagree". On the basis of the "Join Us! Let's Go Green!" program promoted by the Taiwanese Environmental Protection Administration in 2016, the present study defined incentive mechanisms as cash discounts (for accommodation, food, merchandise, and admission tickets for tourist attractions) and eco-friendly substitutes (allocating a portion of funds acquired through consumer environmentally friendly behavior to the sponsorship of green activities). Both cash discounts and eco-friendly substitutes as incentive mechanisms acted as moderator variables (such as high cash discount or low cash discount) rather than exogenous constructs. These two variables were manipulated to alter the magnitude of the causal relationship between DI and BI. In the present study, the collected demographic information included gender, age, marital status, educational attainment, personal monthly income, and the preferred incentive mechanism. The questionnaire items and their reference source are listed in Table 1.

**Table 1.** Constructs or variables and their corresponding statements of measurement included in the questionnaire.

| Construct/Variable | Measuring Items |
| --- | --- |
| Environmental attitude (EA) [55] | EA1: Humans exist to govern nature. <br> EA2: Humans have the right to change nature to suit their needs. <br> EA3: Plants and animals exist to be useful to humans. <br> EA4: Humans do not have to adapt to nature because we can reshape it to suit our own needs. <br> EA5: Our population has almost reached the Earth's capacity. <br> EA6: To sustain natural resources, we must restrict the speed of industrial development. <br> EA7: The balance of nature is delicate and fragile. <br> EA8: When humans interfere with nature, it often produces disastrous consequences. <br> EA9: Humans must strive for harmonic coexistence with nature for survival. <br> EA10: Space and resources on Earth are limited. <br> EA11: Industrial development by economic entities should be limited. <br> EA12: Humans have overly exploited nature. |
| Positive anticipated emotions (PAEs) [37,45] | PAE1: Staying in a green hotel would make me proud of myself. <br> PAE2: Staying in a green hotel would make me happy. <br> PAE3: Staying in a green hotel would make me feel content. <br> PAE4: Staying in a green hotel would make me feel grateful. |

**Table 1.** *Cont.*

| Construct/Variable | Measuring Items |
|---|---|
| Negative anticipated emotions (NAEs) [37,45] | NAE1: Can not staying in a green hotel would make me sad. NAE2: Can not staying in a green hotel would disappoint me. NAE3: Can not staying in a green hotel would make me depressed. NAE4: Can not staying in a green hotel would make me anxious. |
| Perceived behavioral control (PBC) [34,56] | PBC1: I am prepared to stay in a green hotel. PBC2: I have enough money to stay in a green hotel. PBC3: I believe that staying in a green hotel is the right choice. PBC4: I can overcome all obstacles and prioritize staying in green hotels. |
| Subjective norms (SNs) [34,56] | SN1: My friends and relatives support my choice to stay in green hotels. SN2: The pleas of environmental organizations can affect my choices regarding staying in green hotels. SN3: The opinions of renowned experts can affect my choices regarding staying in green hotels. SN4: Promotions by tourism operators can affect my choices regarding staying in green hotels. |
| Desire intentions (DIs) [46] | DI1: During my travels, I want to stay in green hotels. DI2: The likelihood that I will choose to stay in green hotels in the future is very high. DI3: Given the opportunity, I would be willing to stay in green hotels when traveling. |
| Behavioral intention (BI) [34] | BI1: When traveling, I am willing to stay in green hotels. BI2: When traveling, I plan to stay in green hotels. BI3: When traveling, I prefer to stay in green hotels. |
| Incentive mechanisms as cash discounts [30] | IM-cd1: Accommodation discount IM-cd2: Food discount IM-cd3: Merchandise discount IM-cd4: Admission tickets for tourist attractions |
| Incentive mechanisms as eco-friendly substitutes [30] | IM-es1: Allocating a portion of funds acquired through consumer environmentally friendly behavior to the sponsorship of green activities |

### 3.3. Sample Size and Composition

According to the number of eco-friendly hotels in Taiwan as at 2017 [57], there are only 1214 eco-friendly hotels in Taiwan, accounting for 11.38% of the legal hotels and 10,664 qualified hotels (including 3214 legal hotels and 7450 legal hostels). Eco-friendly hotels are not yet popular in Taiwan. This study adopted the convenience sampling method. Most hotel managers are very particular about customer privacy, so only willing owners were selected. Following the initial interview, 10 hotels agreed to cooperate with the survey. Under the condition that customer privacy must be protected and that they must not be disturbed, the survey questionnaire was placed in the room for room guests to fill out, which was then brought out by housekeepers after making up the room. We distributed 80 pilot test survey forms to the hotels between January and March 2017, and 72 of these were returned. After eliminating 22 invalid survey forms, 50 valid pilot test survey forms remained, for a valid return rate of 62.50%. After the pilot test questionnaires were collected, sample coding and data analysis were carried out to verify the reliability and validity of the measurement tools and develop a formal questionnaire. In order to improve the validity of this study, after factor analysis and reliability analysis for 50 valid pilot test forms, DI and BI had one question deleted from the original four questions, one question was split into three questions due to low factor loadings, and the remaining questions remained unchanged.

In total, 500 formal questionnaires were issued to the hotels and 452 responses were retrieved, of which 327 were valid and 125 were invalid, yielding a valid response rate of 65.4%. According to the collected demographic data, 53.6% of the respondents were male and 46.4% were female; the largest age group was 31–40 years (34.6%), followed by 21−30 years (29.7%); most participants were married (57.8%); most were university or community-college graduates (53.3%); the largest group for monthly personal income was NT$35,001–45,000 (31.5%); and most participants preferred the incentive mechanism of cash discounts (57.8%).

### 3.4. Statistical Analysis

This study adopted SEM to examine the TPB. The SEM is an effective model test and improvement method that enables theoretical models to be tested and can explain the causal relationships among the

variables in hypotheses which are related to the models based on statistical dependence. Consumer behavior is often affected by psychological variables that cannot be directly estimated (latent variables), requiring observation variables to be measured indirectly. The SEM can analyze causal relationships between latent variables.

The SEM is divided into two parts: the measurement model and the structural model. The measurement model is the part which relates the measured variables to the latent variables, and factor analysis is used for the evaluation. The structural model is the part that relates latent variables to one another, and path analysis is used for the evaluation. The empirical results are shown in the next section.

## 4. Results

### 4.1. Measurement Model: Reliability and Validity

Construct reliability and validity were quantitatively assessed using the measurement model. Interitem internal consistency was ascertained using Cronbach's $\alpha$, and the score was determined to range from 0.771 to 0.892, which is within the acceptable limit of 0.7 and higher [58]. Discriminant and convergent validity were also determined. Three components were used as the basis for determining the latter: average variance extracted (AVE), composite reliability (CR), and factor loading. The CR value ranged from 0.735 to 0.886, indicating that all constructs met the recommended criterion of demonstrating a CR of 0.6 and higher [59]. The AVE value ranged from 0.673 to 0.806, which also met the acceptable lower limit of 0.5 [58]. However, the value of factor loading (0.706–0.868) was higher than the recommended level of 0.6 [59]. A detailed account of reliability and convergent validity is provided in Table 2.

**Table 2.** Results for factor loading, reliability, and validity.

| Constructs | Items | Factor Loading | Cronbach's $\alpha$ | CR | AVE |
|---|---|---|---|---|---|
| Environmental attitude (EA) | EA1 | 0.817 | 0.862 | 0.838 | 0.729 |
| | EA2 | 0.809 | | | |
| | EA3 | 0.842 | | | |
| | EA4 | 0.787 | | | |
| | EA5 | 0.793 | | | |
| | EA6 | 0.812 | | | |
| | EA7 | 0.829 | | | |
| | EA7 | 0.794 | | | |
| | EA9 | 0.816 | | | |
| | EA10 | 0.851 | | | |
| | EA11 | 0.787 | | | |
| | EA12 | 0.826 | | | |
| Positive anticipated emotions (PAEs) | PAE1 | 0.804 | 0.874 | 0.871 | 0.764 |
| | PAE 2 | 0.836 | | | |
| | PAE 3 | 0.794 | | | |
| | PAE 4 | 0.815 | | | |
| Negative anticipated emotions (NAEs) | NAE1 | 0.712 | 0.838 | 0.852 | 0.697 |
| | NAE 2 | 0.836 | | | |
| | NAE 3 | 0.762 | | | |
| | NAE 4 | 0.815 | | | |
| Perceived behavioral control (PBC) | PBC1 | 0.762 | 0.771 | 0.735 | 0.673 |
| | PBC2 | 0.706 | | | |
| | PBC3 | 0.802 | | | |
| | PBC4 | 0.868 | | | |
| Subjective norms (SNs) | SN1 | 0.772 | 0.837 | 0.863 | 0.794 |
| | SN 2 | 0.861 | | | |
| | SN 3 | 0.816 | | | |
| | SN 4 | 0.826 | | | |
| Desire intention (DI) | DI1 | 0.832 | 0.892 | 0.886 | 0.745 |
| | DI2 | 0.786 | | | |
| | DI3 | 0.853 | | | |
| Behavioral intention (BI) | BI1 | 0.778 | 0.824 | 0.764 | 0.806 |
| | BI2 | 0.827 | | | |
| | BI3 | 0.858 | | | |

Note: CR: Composite reliability; AVE: Average variance extracted.

### 4.2. Interrelationships between Variables and Goodness-of-Fit

Means, standard deviations, and correlations among constructs are presented in Table 3. Significant positive correlations were found to exist between EA and DI (r = 0.646, *p* < 0.01), PAEs and DI (r = 0.870, *p* < 0.01), SNs and DI (r = 0.347, *p* < 0.01), and PBC and DI (r = 0.501, *p* < 0.01). These results indicate that the higher the EA, PAEs, SNs, and PBC of travelers, the stronger their DI to stay in green hotels. By contrast, NAEs exhibited a significant negative correlation with DI (r = −0.725, *p* < 0.01), indicating that the higher the NAEs of travelers, the weaker their DI to stay in green hotels. Furthermore, DI and BIs exhibited a significant positive correlation (r = 0.876, *p* < 0.01), indicating that the stronger the DIs of travelers to stay in green hotels, the greater their BIs to do so.

**Table 3.** Correlations between variables.

| Variable | Mean | Standard Deviation | 1 | 2 | 3 | 4 | 5 | 6 | 7 |
|---|---|---|---|---|---|---|---|---|---|
| 1. EA | 6.028 | 0.667 | 1.000 | | | | | | |
| 2. PAEs | 5.364 | 1.127 | 0.414 ** | 1.000 | | | | | |
| 3. NAEs | 3.197 | 1.248 | −0.314 ** | −0.357 ** | 1.000 | | | | |
| 4. SNs | 5.017 | 1.039 | 0.448 ** | 0.489 ** | 0.350 ** | 1.000 | | | |
| 5. PBC | 4.691 | 1.036 | 0.362 ** | 0.605 ** | 0.343 ** | 0.452 ** | 1.000 | | |
| 6. DI | 5.307 | 1.079 | 0.646 ** | 0.870 ** | −0.725 ** | 0.347 ** | 0.501 ** | 1.000 | |
| 7. BI | 5.374 | 1.038 | 0.571** | 0.672** | −0.281 ** | 0.772 ** | 0.439 ** | 0.876 ** | 1.000 |

Note: ** *p* < 0.01.

A goodness-of-fit test conducted on the theoretical framework yielded the following results, which lie within the acceptable limits: $x^2/df$ = 1.475, goodness-of-fit index (GFI) = 0.875, root mean square error of approximation (RMSEA) = 0.024, standardized root mean square residual (SRMR) = 0.036; adjusted GFI (ACFI) = 0.849; normalized fit index (NFI) = 0.906; comparative fit index (CFI) = 0.965; and parsimonious normed fit index (PNFI) = 0.823. All other fit indices were above the recommended criteria. As a result, all indices provided evidence of an acceptable measurement model (Table 4).

**Table 4.** Results of the goodness-of-fit indicators for the evaluation model.

| Fit Index | Ideal Value | Result | Conclusion |
|---|---|---|---|
| $\chi^2/\mathrm{df}$ | <3 | 1.475 | Acceptable |
| GFI | >0.9 (good fit) 0.8–0.89 (acceptable fit) | 0.875 | Acceptable |
| AGFI | >0.9 (good fit) 0.8–0.89 (acceptable fit) | 0.849 | Acceptable |
| NFI | >0.9 | 0.906 | Acceptable |
| CFI | >0.9 | 0.965 | Acceptable |
| PNFI | >0.5 | 0.823 | Acceptable |
| SRMR | <0.05 | 0.036 | Acceptable |
| RMSEA | ≤0.05 (close fit) 0.05–0.08 (fair fit) 0.08–0.10 (mediocre fit) >0.10 (poor fit) | 0.024 | Close fit |

Note: GFI: goodness-of-fit index; AGFI: adjusted goodness-of-fit index; NFI: normalized fit index; CFI: comparative fit index; PNFI: parsimonious normed fit index; SRMR: standardized root mean square residual; RMSEA: root mean square error of approximation.

### 4.3. Structural Model and Hypothesis Testing

After achieving a better fit of the measurement model (Table 4), SEM was further performed using the maximum likelihood estimation method to evaluate the model proposed in this paper. Specifically, SEM was used to test nine cause-and-effect hypotheses among seven latent constructs (including three exogenous constructs: EA, SNs, and PBC, and two endogenous constructs: DI and BI) and two moderating variables (cash discounts and eco-friendly substitutes). This study assesses factor invariance of the measurement when manipulating multigroup comparisons. The results of measurement invariance across groups indicated the full invariance across samples. The overall goodness-of-fit statistics for the structural model also showed a moderate fit of the data to the model (such as CFI = 0.962, SRMR = 0.041, and RMSEA = 0.030). It was established that it could predict outcome variables satisfactorily. Furthermore, a multiple group analysis within Amos was adopted to estimate the moderating effects of both cash discounts and eco-friendly substitutes on the structural model by testing the chi-square value difference. Specifically, the chi-square value for the unconstrained and the constrained models were statistically significant at the level of $\alpha$ = 0.01 ($p$ < 0.001), indicating that both cash discounts and eco-friendly substitutes had a moderating effect on the structural model. Table 5 indicates that the effect of DIs and BIs was stronger in the cash discounts group ($\beta$ = 0.857 ***) than the effect in the eco-friendly substitutes group (0.648 ***). Results of the path analysis and verification of hypotheses including two moderating effects are presented in Table 5 and Figure 2.

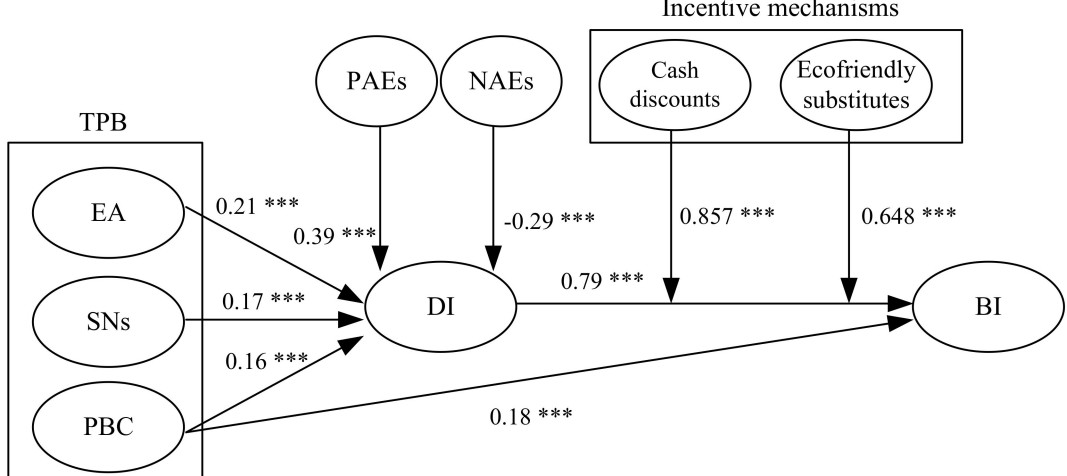

**Figure 2.** Paths within the hypothesis model. Solid lines denote established hypotheses (note: *** $p < 0.001$).

**Table 5.** Path analysis results and hypothesis verification.

| Hypothesis | Hypothesized Path | Path Coefficient | Results |
|---|---|---|---|
| H1 | EA→DI | 0.21 *** | Supported |
| H2 | PAEs→DI | 0.39 *** | Supported |
| H3 | NAEs→DI | −0.29 *** | Supported |
| H4 | SNs→DI | 0.17 *** | Supported |
| H5 | DI→BI | 0.79 *** | Supported |
| H6 | PBC→DI | 0.16 *** | Supported |
| H7 | PBC→BI | 0.18 *** | Supported |
| | Hypothesized moderated path | Path coefficient | Results |
| H8a | Cash discounts moderate: DI→BI | 0.857 *** | Moderation verified |
| H8b | Ecofriendly substitutes moderate: DI→BI | 0.648 *** | Moderation verified |

Note: *** $p < 0.001$.

## 5. Discussion and Implication

First, the results suggest that EA had a significant positive effect on the DI to stay in green hotels. This echoes the findings reported by Chan and Hsu [35] in that greater consumer EAs correspond to stronger DIs to stay in green hotels. Likewise, the SNs and PBC of consumers were revealed to have significant positive effects on their DI, which is in agreement with the findings of Yadav and Pathak [60] in that when the parents, spouses, and peers of a consumer assume positive attitudes toward green hotels, the consumer's own DI to stay in green hotels increases, thereby enhancing their corresponding BIs. These results suggest that when consumers consider whether to stay in green hotels, their DI can be influenced by their EA, PAEs, SNs, and PBC. Second, the results suggest that the NAEs of consumers have significant negative effects on their DI to stay in green hotels; in other words, lower consumer NAEs correspond to higher consumer DIs. Therefore, reductions in NAEs can be inferred to effectively enhance the DIs of consumers to stay in green hotels. This is in agreement with the findings of Han and Yoon [1], who concluded that because reducing the NAEs of consumers can effectively raise their DI to stay in green hotels, awareness of a hotel's environmental friendliness

should be promoted to reduce the mistrust and, consequently, the NAEs of consumers. In addition, the PBC of consumers exhibited a significant positive effect on their BIs, echoing the findings reported by Litvine and Wüstenhagen [48] that consumers tend to stay in green hotels when traveling if they perceive that they have the time and ability to do so. Moreover, the analysis revealed a positive correlation between DI and BIs, which agrees with the findings of a study conducted by Han and Yoon [1] in that a consumer's BIs toward staying in green hotels increase with their DI to do so. Thus, H5 is supported. Last, the moderating effects of two incentive mechanisms were examined in this study: cash discounts and eco-friendly substitutes. The results indicated that consumers favored cash discounts over eco-friendly substitutes. This finding is consistent with the results of a study conducted by Frey and Jegen [51], who reported that cash returns more significantly increased consumer DI to stay in green hotels compared with eco-friendly substitutes.

This research provides several implications for hoteliers. This study applied the TPB to explore the behavioral intentions of customers staying in green hotels. According to the research results, "Space and resources on Earth are limited", "Plants and animals exist to be useful to humans", and "Humans have overly exploited nature" are the top three factors that are most likely to explain EA. In the "SNs" dimension, "The pleas of environmental organizations can affect my choices regarding staying in green hotels", "The opinion of renowned experts can affect my choices regarding staying in green hotels", and "Promotions by tourism operators can affect my choices regarding staying in green hotels" are the top three factors most able to explain SNs; in the "PBC" dimension, "I can overcome all obstacles and prioritize staying in green hotels" and "I believe that staying in a green hotel is the right choice" are the two factors most likely to explain the PBC. Therefore, it is recommended that the hoteliers emphasize two key points in promoting the marketing of their "green hotel": reducing ecological damage and strengthening the promotion of green education.

Firstly, to reduce ecological damage, the following should be considered: (1) implementing a green procurement plan, and environmental product procurement should follow environmental protection labels or water-saving labels and so on; (2) reducing disposable products (e.g., disposable toiletries, such as small packages of shampoo, shower gel, soap, toothpaste, and toothbrushes) and reducing waste; (3) lessening the impact of disposable products on the environment; (4) implementing hazardous substance management; for example, increasing the number of rechargeable batteries used and avoiding the use of halogen-based solvents as cleaning agents; (5) implementing waste separation and resource recovery; for instance, the housekeeping managers can formulate a plan to reduce the amount of garbage in the hotel guestrooms, and the room maids can be trained to sort, divide, and recycle the available resources when making up the room, and thus minimize the amount of garbage in the hotel.

Secondly, we discuss strengthening the promotion of green education. "Environmental protection work" promotes the concept of environmental protection. At present, most tourists are not yet familiar with the concept of hotels reducing disposable products and reducing the frequency of sheet and towel replacement. Some of the hotels have taken the initiative to cooperate with the EPD policy to implement environmental protection measures and do not to offer room guests disposable products, such as toothpaste, toothbrushes, razors, and combs. Moreover, the hotelier will place a card about saving energy and water resources and reducing waste next to the in-room washstand to educate consumers on reducing the use of disposable toiletries. If the hotel guest still needs toothpaste and a toothbrush, the receptionists will offer them the "disposable toiletries pack" and remind room guests to bring their own personal toiletries on the next trip. As the awareness of environmental protection rises, it is more important to educate hotel guests about environmental protection during their stay.

The transformation of the hotel industry into green hotels is at the sacrifice of convenience and the quality of service. Although habits related to self-prepared personal toiletries are different when traveling, the green hotel is the future trend. The hotel industry promotes the "green hotel", improving the overall image of hotels and potentially reducing operating costs. By delivering green education to

hotel guests and imperceptibly influencing their behavior during their stay, it will help the hoteliers of green hotels achieve sustainable management practices.

## 6. Conclusions

### 6.1. Findings

This study finds that the EA of consumers exhibited a significant positive effect on their DI to stay in green hotels; that is, after consumers become aware of the critical state of the environment and the severity of its destruction and start to regard environmental protection as imperative, they tend to select green hotels. Furthermore, consumer SNs demonstrated a significant positive effect on this DI, which indicates that if a consumer's peers possess deeper environmental awareness or assume a positive attitude toward green hotels, their DI to select green hotels increases. Consumer PAEs also exhibited a significant positive effect on the DI to stay in green hotels; this implies that if consumers presume that they will not be disappointed by green hotels and if their actual experience of staying in such hotels is satisfactory, then their DI to opt for green hotels will increase. However, consumer NAEs demonstrated a significant negative effect on this DI; this means that if consumers expect to both feel anxious when staying in green hotels and regret the decision and if their actual experience of staying in such hotels is also negative, then their DI to choose green hotels will decrease. A consumer's PBC also exhibited a significant positive effect on their DI to stay in green hotels, indicating that if consumers perceive that they have the time and means to fulfil their intention of staying in green hotels, their DI to do so may increase.

Studies on green hotels have tended to emphasize how demographic characteristics (gender, age, socioeconomic status, and educational attainment) affect BIs. However, devising various marketing strategies for guests with different demographic characteristics is impractical for hotels. The results of the present study can serve as a valuable reference for the hotel industry because they reveal that consumers with different demographic characteristics consider cash discounts attractive when opting to stay in green hotels. In other words, businesses in the hotel industry should be advised to adopt cash discounts in their promotional strategies.

In recent years, environmental sustainability has received attention [61]. Hotel management is becoming more concerned regarding environmental issues. Consumers are aware that their purchasing behavior is related to environmental issues. Han and Yoon [1] and Chen and Tung [23] studied the behavioral intentions of consumers to visit green hotels. The empirical results of Han et al. [1] show that attitude, subjective norm, and perceived behavioral control positively affected intention to stay at a green hotel. Han and Yoon [1] pointed out environmental behaviors such as recycling or purchasing environmentally friendly products are not related to green hotel purchasing decisions. This shows that if hotel management only emphasize the importance of environmental protection issues, it is not enough to attract customers to visit green hotels. Further, Chen and Tung [23] proved that consumers' environmental concern have a positive attitude influence on green hotels. However, insufficient information can affect green consumer' behavioral intentions, and hotel management cannot effectively communicate the hotel's environmental policies and business philosophy to consumers [23]. Tsai and Tsai [62] suggested that hotels can implement discounts for not requesting new towels daily or for not using hotel-provided hygiene and sanitation products. Therefore, this present study sought to explored the incentive mechanisms that influence consumers regarding the decision to visit green hotels, and the result shows that cash discounts and eco-friendly substitutes can attract consumers to choose green hotels.

### 6.2. Study Limitations and Scope for Future Research

This piece of research is limited in several ways. First, instead of asking the respondents to describe their opinions about a specific hotel type, the researchers asked them to provide general information about their perspectives concerning various hotel attributes, as well as their BIs in response to green

hotels. Similarly, consumer expectations and behavior may differ across green hotel categories; thus, future studies on staying in green hotels can offer valuable extra insight if they study domain-specific BIs and attitudes.

It is recommended that future research distinguishes three levels of service in eco-friendly hotels, such as world class service (e.g., Luxury/ Five Start hotels), mid-range service (e.g., 3 to 4-start hotels), and budget hotel with limited service (e.g., budget hotels), and stratifies the samples to distribute the samples evenly. In addition, the *independent variables* of this study are individual psychological factors. It is suggested that future research on eco-friendly hotels should further explore the factors that hinder travelers from choosing an eco-friendly hotel (e.g., inconvenience and travel motivation), use contextual variables (such as employer brands, etc.), and explore other "perception" variables (such as perceived sacrifice, perceived risk, perceived inconvenience, etc.) to more fully clarify consumer behavior.

**Author Contributions:** The four coauthors together contributed to the completion of this article. C.-T.T. was the first author, who analyzed the data and drafted the manuscript; C.-M.H. contributed to reviewing the manuscript and revising the results and conclusion; H.-P.C. contributed to reviewing and revising the literature, results, and conclusion; and H.-S.C. acted as the corresponding author on their behalf throughout the revision and submission process.

**Funding:** This research received no external funding.

**Conflicts of Interest:** The authors declare no conflict of interest.

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
