# Peer review of "Environmental Consciousness and Green Customer Behavior: The Moderating Roles of Incentive Mechanisms"

_sustainability, doi:10.3390/su11030819_

Round 1

Reviewer 1 Report

I appreciate the opportunity to review your paper and contribute to the improvement of your research work. I hope you will find my comments constructive to improve the quality of your research. 

The objective of this paper is  twofold: to explore the main reasons that consumers opt to stay in green hotels and determine which incentive mechanism (cash discounts or eco-friendly substitutes) is more effective in enhancing the BIs 66 of consumers to stay in green hotels.

Improve literature review with following work:

For TPB:

Chuang, L.-M., Chen, P.-C., & Chen, Y.-Y. (2018). The Determinant Factors of Travelers’ Choices for Pro-Environment Behavioral Intention-Integration Theory of Planned Behavior, Unified Theory of Acceptance, and Use of Technology 2 and Sustainability Values. Sustainability, 10(6), 1869. http://doi.org/10.3390/su10061869

Reyes-Menendez, A., Saura, J., Palos-Sanchez, P., & Alvarez-Garcia, J. (2018). Understanding User Behavioral Intention to Adopt a Search Engine that Promotes Sustainable Water Management. Symmetry, 10(11), 584. http://doi.org/10.3390/sym10110584

For sustainable management:

Reyes-Menendez, A., Saura, J., & Alvarez-Alonso, C. (2018). Understanding #WorldEnvironmentDay User Opinions in Twitter: A Topic-Based Sentiment Analysis Approach. International Journal of Environmental Research and Public Health, 15(11), 2537. http://doi.org/10.3390/ijerph15112537

Authors have to justify each construct individually (L172-173, L192-193, L209-211)

Improve the justification of hypothesis with recommended literature improvement and more

Table 1 (L236) is difficult to follow as it is too long, please reorganize the spaces of the cells

Take the implications outside of the conclusions and create a new section prior to the conclusions

Reviewer 2 Report

The topic of the paper is interesting as well as the academic contribution of the work.

The article reflects the present state of knowledge. The text is easy to understand by scientists in other disciplines.

The paper is clear, well written and well organized. The methodological approach is technically correct.

Conclusions should be rewritten to understand the importance of research. The conclusion section should highlight more clearly how the results compare with the recent literature review.
